# Assessing the importance of demographic risk factors across two waves of SARS-CoV-2 using fine-scale case data

**Anthony J. Wood**[1], **Aeron R. Sanchez**[1], **Paul R. Bessell**[1], **Rebecca Wightman**[2], **Rowland R. Kao**[1,3]*

**1** Roslin Institute, University of Edinburgh, Midlothian, United Kingdom, **2** Edinburgh Medical School, University of Edinburgh, Edinburgh, United Kingdom, **3** Royal (Dick) School of Veterinary Studies, University of Edinburgh, Midlothian, United Kingdom

* rowland.kao@ed.ac.uk

**Data Availability Statement:** Analysis code is available at https://git.ecdf.ed.ac.uk/awood310/scotland-covid-case-distribution-random-forest-model. The COVID-19 testing, vaccination and

## Abstract

For the long term control of an infectious disease such as COVID-19, it is crucial to identify the most likely individuals to become infected and the role that differences in demographic characteristics play in the observed patterns of infection. As high-volume surveillance winds down, testing data from earlier periods are invaluable for studying risk factors for infection in detail. Observed changes in time during these periods may then inform how stable the pattern will be in the long term. To this end we analyse the distribution of cases of COVID-19 across Scotland in 2021, where the location (census areas of order 500–1,000 residents) and reporting date of cases are known. We consider over 450,000 individually recorded cases, in two infection waves triggered by different lineages: B.1.1.529 ("Omicron") and B.1.617.2 ("Delta"). We use random forests, informed by measures of geography, demography, testing and vaccination. We show that the distributions are only adequately explained when considering multiple explanatory variables, implying that case heterogeneity arose from a combination of individual behaviour, immunity, and testing frequency. Despite differences in virus lineage, time of year, and interventions in place, we find the risk factors remained broadly consistent between the two waves. Many of the observed smaller differences could be reasonably explained by changes in control measures.

## Author summary

The COVID-19 pandemic has seen unprecedented amounts of high-quality data collected for a human disease. For longer-term control in the absence of widespread testing, these data are invaluable for understanding whom amongst the population is at the highest risk of infection. In this work we fit the detailed distributions of COVID-19 cases over Scotland, across two infection waves driven by different variants, to identify risk factors. These were at a time when Scotland had substantial population immunity from prior infection and vaccination, and strict control measures were being relaxed. Differences across the waves may then indicate how stable the pattern of infection will be in the longer term.

hospitalisation data utilised in this work are not publicly available. They are provided to the authors for academic research by Public Health Scotland's electronic Data Research and Innovation Service, under a data sharing agreement ("Spatial and Network Analysis of SARS-CoV-2 Sequences to Inform COVID-19 Control in Scotland"), and can be contacted via phs.edris@phs.scot. The authors received no special privileges with respect to data access as compared to other researchers. Data relating to deprivation are derived from the 2020 Scottish Index of Multiple Deprivation, and are publicly available (https://simd.scot). Fine-scale population statistics are drawn from publicly-available population estimates as of mid-2020 (https://www.nrscotland.gov.uk/statistics-and-data/statistics/statistics-by-theme/population/population-estimates). The population of students and those belonging to a minority ethnicity are drawn from public Scottish census data (tables KS501SC, LC2101SC respectively).

**Funding:** This work has been funded by the ESRC grant "Real-time monitoring and predictive modelling of the impact of human behaviour and vaccine characteristics on COVID-19 vaccination in Scotland" (R.R.K., ES/W001489/1). R.R.K. has also been supported through the BBSRC Institute Strategic Programme grant to the Roslin Institute (BB/J004235/1). The funders had no role in study design, data collection and analysis, decision to publish, or preparation of the manuscript.

**Competing interests:** The authors have declared that no competing interests exist.

Despite Scotland's high geographic and demographic diversity, we effectively fit the case distribution in both waves, and find only minor variation between the two. Uniquely, our model was informed by the volume of *negative* COVID-19 lateral flow tests, and we find that a high rate of negative test reporting was a risk factor for a high rate of cases. This, combined with high variability in testing across demographics, leads us to suggest that patterns in reported case data may in fact be quite different to those of all infections, reported and unreported.

## 1 Introduction

A key challenge in the long term control of an infectious disease is to identify predictable patterns of incidence. The emergence and spread of the SARS-CoV-2 virus saw restrictions imposed globally on everyday life to control the spread of COVID-19 infection, and to protect individuals at highest risk of severe disease. While as of March 2023 few to no restrictions remain in place in Scotland, as in the rest of the UK, randomised testing [1] and hospital admissions [2] indicate continued widespread transmission. The winding down of community testing and other surveillance is making it more difficult to track the transmission patterns of COVID-19 in detail.

Typically, identifying risk factors for infection rely on disease surveillance studies. While these studies can be powerful and provide important insights [3–6], they are often expensive, laborious and time consuming. "Big Data" in the health sciences offers an opportunity to gain some of the same insights using routinely collected data. The availability of COVID-19 case data at fine spatial scales with detailed metadata enables us to identify important health-related risks, with the data collected during the pandemic being made available to researchers in close to real-time.

In this work we aim to identify risk factors for COVID-19 cases in Scotland, and their change over time, to serve as an indicator for how the longer-term profile of infection may evolve. We fit the case distributions of two different waves of COVID-19, with a machine learning model informed by a range of explanatory variables relating to geography and demographics.

The first COVID-19 case in Scotland was identified on 1$^{st}$ March 2020 [7]. The Scottish Government imposed strict "lockdown" non-pharmaceutical intervenions (NPIs) on 23$^{rd}$ March 2020 [8]. While initially applied at the national level, following the initial lockdown period NPIs were adjusted by local authority (administrative areas with populations ranging between 22,540–635,130) through a "levels"-based system [9]. The seeding and rapid spread of the B.1.1.7 lineage (termed the "Alpha" variant) in December 2020 led to a tightening of NPIs and a second lockdown [10, 11]. A mass vaccination programme began in December 2020 [12, 13], prioritising the elderly and healthare workers, with all adults eventually eligible.

We focus on case data gathered between May 2021 and January 2022, a period that saw the steady relaxation of nearly all NPIs [14]. This period had two major waves of infection: the first from May 2021 triggered by the B.1.617.2 lineage ("Delta"), and a second wave from November 2021 by the B.1.1.529 B.A.1 lineage ("Omicron"). The deletion of two specific amino acids in the Omicron sub-variant distinguished it from most co-circulating variants including Delta, in PCR tests that have an accompanying "S-gene" test result [15]. A high-capacity testing programme was in place throughout, with free-of-charge lateral flow testing strongly encouraged, and PCR testing mandated for those with symptoms, or a lateral flow positive.

Earlier work has exploited finely-grained case data to highlight risk factors for cases and severe outcomes including (but not limited to) sex [16–18], population density [19–21], deprivation [22–25], occupation [26–28], and age [29–31]. Similar studies have incorporated movement data [32] to demonstrate the protective impact of NPIs that restrict mobility [21, 33–37]. Many of these studies focus on the "first wave" of infection, during which strict NPIs were imposed and no population immunity had been established. This study focuses on a more advanced period moving away from NPIs, and the conditions for disease spread comparatively less "exceptional". This is especially the case for the Omicron wave. A unique feature of our model is the inclusion of lateral flow test taking *frequency*. The proportion of infectons that end up reported is likely to depend on testing propensity, and we consider how that may lead to distortions in the case distribution.

Our main finding is that the risk factors for cases remained broadly consistent across both waves. Differences between the two waves either offer relatively small scale changes in demographic risk or are consistent with the impact of changes in approaches to control.

## 2 Results

The period November 15[th] 2021—January 6[th] 2022 covers the first outbreak and peak of the B.1.1.529 lineage (BA.1 sublineage, hereafter referred to as the Omicron variant) (S-gene "dropout" test signature). Prior to this, the B.1.617.2 lineage (Delta variant) (S-gene positive test signature) was dominant. From 15[th] November 2021, S-gene dropout cases consistently rise, and all subsequent "dropout" cases are assumed Omicron. Remaining S-gene positive cases are presumed to be Delta, consistent with nationwide sequence data [38].

### 2.1 Time evolution and early patterns of spread

We identified 385,558 cases between November 15[th] 2021 and January 6[th] 2022, of which 227,286 were likely Omicron. From 1[st] May 2021 to 7[th] September 2021 we identified 269,838 cases, of which 229,073 were likely Delta. The remaining cases in these periods (those with no S-gene result, or a different result) are excluded. The start date for each of these periods is the first date from which there are consistent rises in cases that are likely the new variant.

Omicron cases had a doubling time (the time taken for *newly reported daily* cases to double) of 2.9 days over the first 28 days, compared to 6.2 days for Delta (Fig A in S1 Text). Over half of all DZs had reported an Omicron case in the wave within 29 days, whereas for Delta this took 39 days (Fig B in S1 Text).

The reproduction number $R_t$ consistently rose for Omicron, peaking at above 2 for nearly all local authorities 28 days in to the outbreak, and only consistently falling below 1 after 50 days (Fig C in S1 Text). Reproduction numbers for Delta are less consistent between LAs; while the number generally remains above 1 for most LAs in the period, there is no coherent peak at the start of the wave.

In the intermediate period during which Omicron became dominant and Delta declined, the *age* distributions by variant differed (Fig D in S1 Text). Taking the mid-points of the five-year age brackets, the mean ages of the Delta-type cases was 3.9 years lower than the Omicron-type cases (31.8 years compared to 35.7 years). A Student's $t$ test shows this difference to be statistically significant ($t = -52.2$, $p < 0.001$). This was the case from relatively early on when Omicron accounted for at least 5% of cases. However, the median ages are equal (both 32.5 years), as in the Omicron-type cases there is a trough in those aged 0–14, with fewer than 50% of cases in this age group Omicron, but then a peak in the 20–29 age group.

## 2.2 Case distribution and model fit

Fig 1, shows the distribution of COVID-19 cases for the Omicron and Delta waves broken down by age, sex, prior cases (serving as a proxy for prior immunity from infection), deprivation and health board. Omicron case rates were highest in younger adults, peaking at 90 cases/

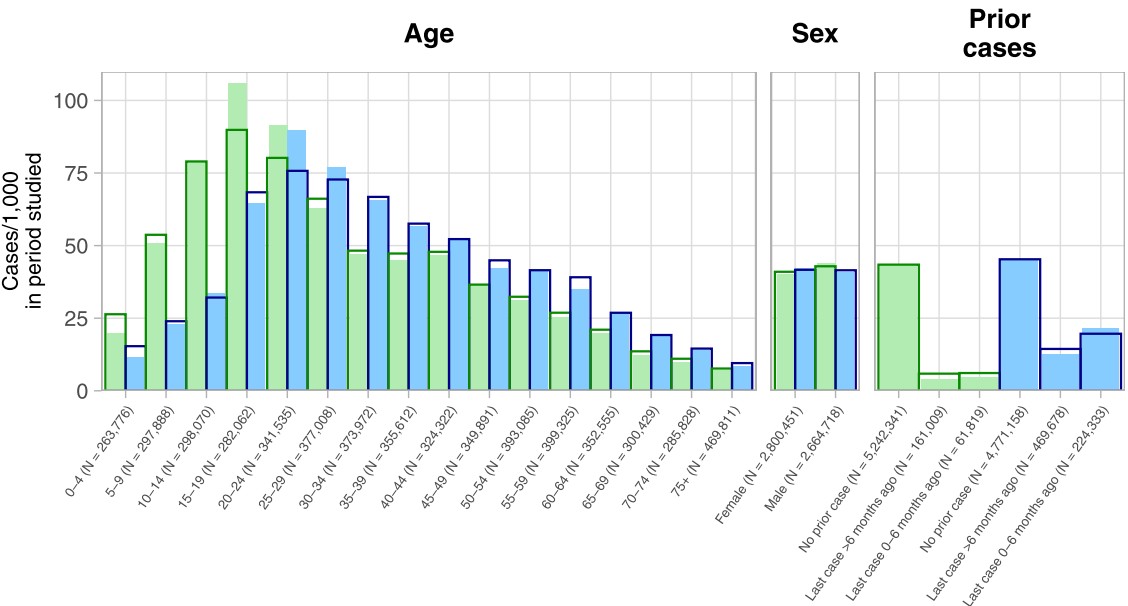

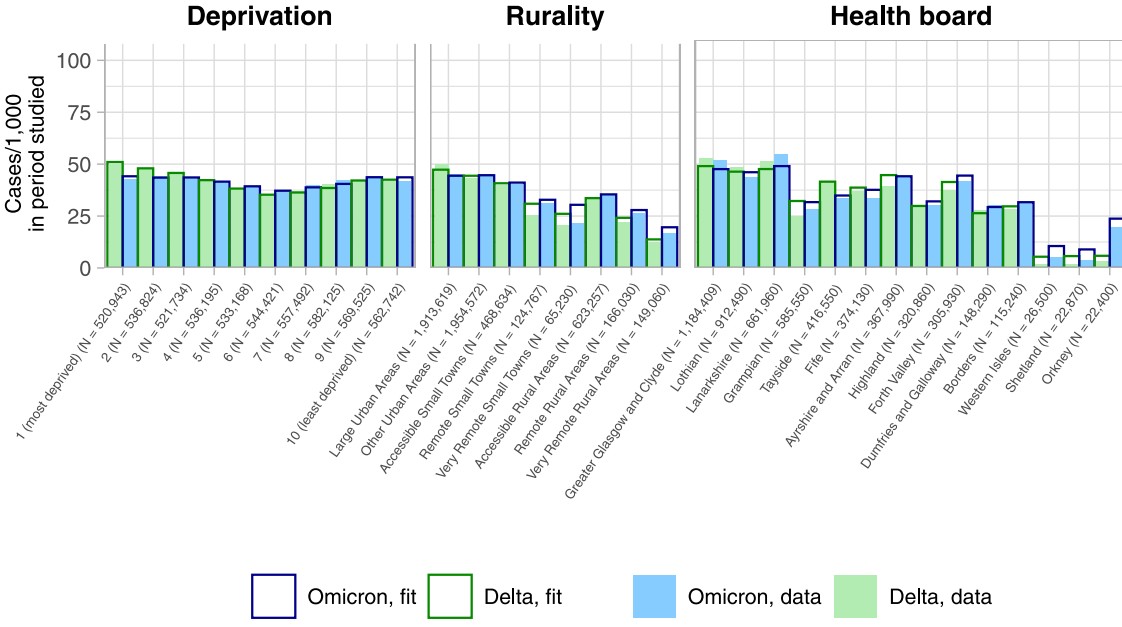

**Fig 1.** Summary of 227,286 Omicron COVID-19 cases in Scotland between November 15th 2021 and January 6th 2022 (blue, filled), and 229,073 Delta cases from 1st May 2021 to 7th September 2021 (green, filled). The full population (*N* = 5, 465, 169) is broken down by *age range*, *prior case status* (whether a person had previously reported a COVID-19 case prior to that specific wave, and when), *deprivation* (of place of residence, per the SIMD decile, with 1 the most deprived), *rurality* (of place of residence, per the census Urban/Rural Classification) and *location* (at the level of Scottish health board). Cases are given per 1,000 people in that group (with subpopulation *N* recorded on the axis labels). The corresponding case rates as fit by our models are superimposed. Note that the subpopulations in the *prior case status* plot change across waves, due to being at different points in time.

1,000 in ages 20–24. There was only a small difference in rates between men and women. Case rates were much lower amongst those that had tested positive for COVID-19 previously. Fig 2 shows case rates per DZ. Geographically, case rates fall with increasing rurality, most notably in Orkney, Shetland and the Western Isles (all island communities). The trend with respect to

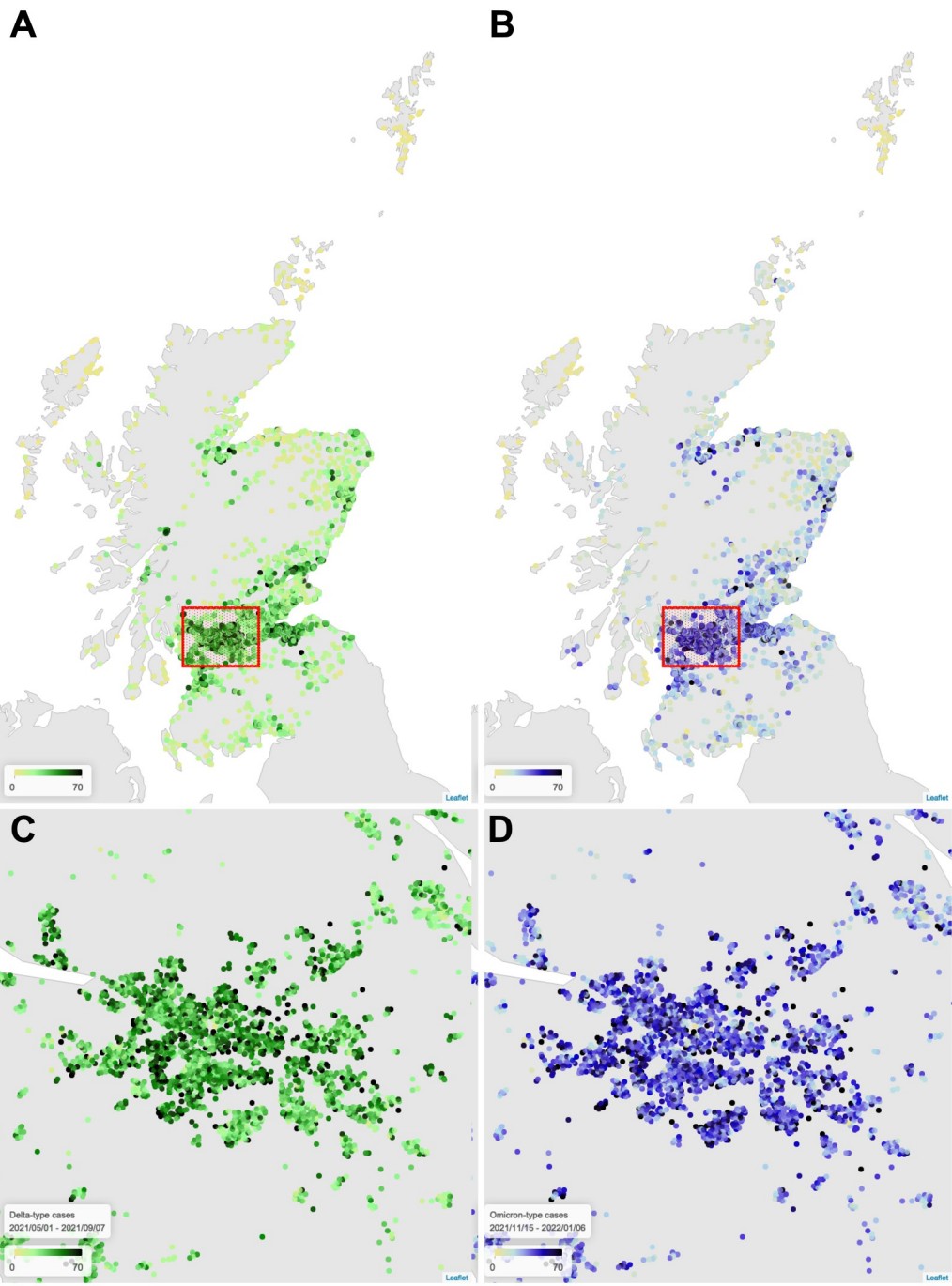

**Fig 2. COVID-19 cases in Scotland over the Delta period (A) as compared to Omicron (B), with focus on the Greater Glasgow region (C, D).** Each point indicates the population-weighted *centroid* of a DZ, with the colour representing the number of cases reported. Base maps obtained from Natural Earth [39].

multiple deprivation decile is bimodal, with higher rates towards the highest and lowest deciles.

The fit case rates from our random forest regression models are overlaid onto Fig 1. We achieve a good fit to these larger-scale trends. The model slightly under-fits the age ranges 15–24, where case rates were the highest overall. Variable importance outputs are presented in Fig G in S1 Text, with node purity and accuracy loss.

Fig 3A shows model performance at DZ level, comparing observed cases to fit cases. Beginning with Omicron cases, our full model explains 70% (fit: 71%, test: 62%) of local variation in the case distribution (R-squared for case numbers, aggregated at a DZ level), with a poorer fit

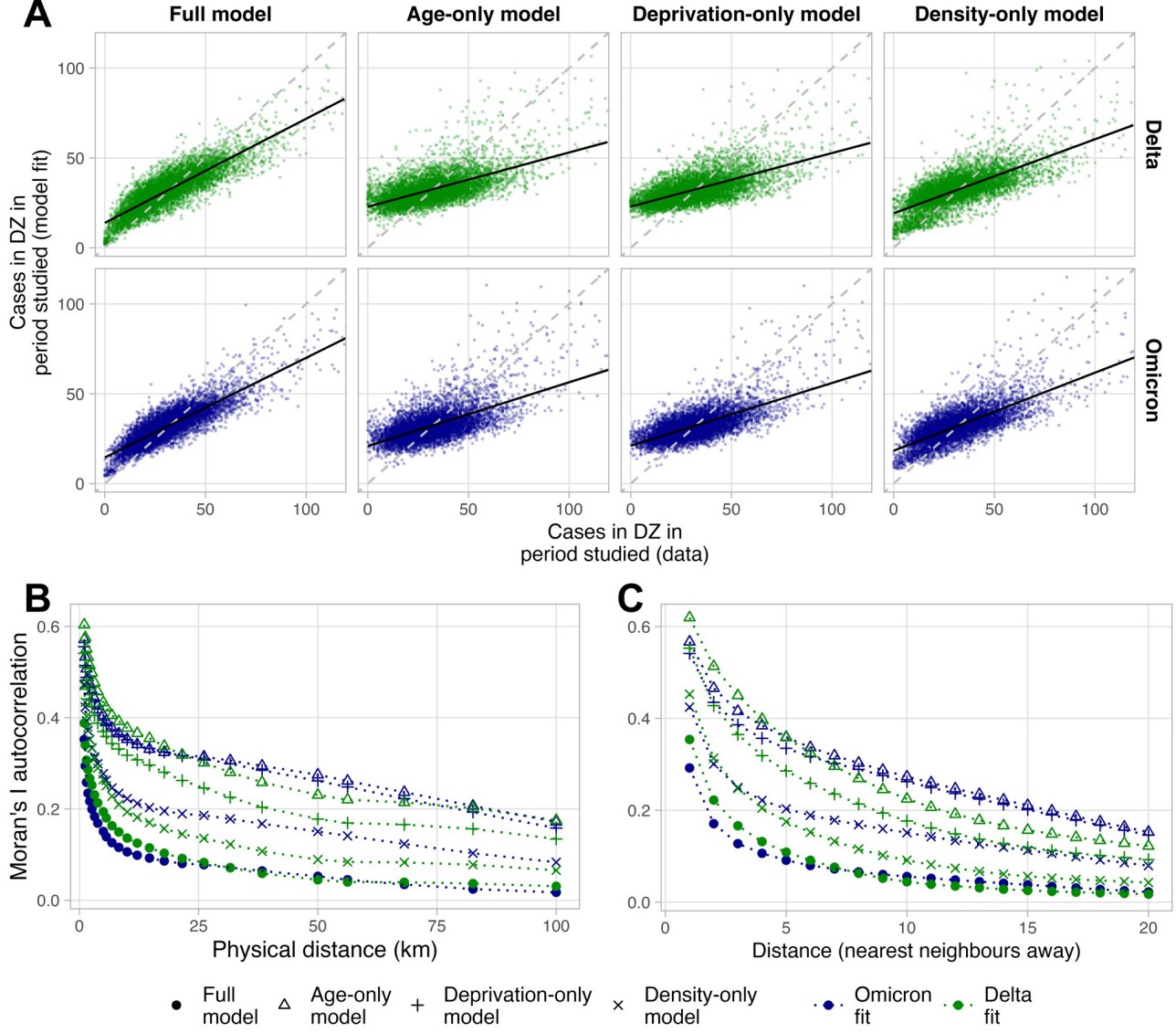

**Fig 3. Performance of different models.** (A) comparing observed cases to fit cases at DZ level. Each point represents a DZ. Points deviating from the diagonal indicate DZs with less accurate fits. The full model is compared with performance of reduced models informed with only population, and one of either age, overall deprivation rank, or population density. Also shown is residual clustering as measured by the Moran's I statistic, at different physical distances (B) and network-based distances (C). Higher values represent higher autocorrelation between model residuals, when comparing DZs sitting within a given locus. DZs are defined as nearest neighbours of one another if they share a boundary.

for cohorts with very high case counts. A "reduced" random forest model informed by population and population density alone explained 59% (fit: 60%, test: 55%) of variation. A model informed by only population/deprivation rank explained 53% (fit: 53%, test: 51%), and one informed by only population/age explained 48% (fit: 48%, test: 51%). Fig 3A shows further deviation of the data-fit slopes away from the diagonal for these "reduced" models.

Considering now earlier Delta cases from 1st May to 7th September 2021, the geographical distribution (Fig 2) is visually similar, with a concentration of high case rates in the denser "central belt". Cases skewed slightly younger (Fig 1), with the highest rates within ages 15–19. The distributon with respect to deprivation decile remains bimodal, with higher rates in both the most and least deprived DZs. Model performance was similar, explaining 72% (fit: 73%, test: 61%) of DZ-level variation.

Fig 3B and 3C shows for both the Delta and Omicron models, autocorrelation of residuals (as measured by the Moran's I statistic, Section 4.5) within 1km is 0.35, falling to 0.15 at 5km, and 0.05 at 50km. The reduced models exhibit much higher residual autocorrelation, with the density-only model performing best, but persisting over larger distances (see Fig F in S1 Text for a map view of residuals).

## 2.3 Accumulated local effects

Fig 4 shows the accumulated local effects (ALEs) of all explanatory variables in the model (see Section 4.4 for definition).

Population, age, sex, and prior case status have ALEs that follow the empirical distributions observed in Fig 1; ALEs are strongly positive for ages between 15–40, and those that had never reported a case before.

Beyond these variables, Fig 4 shows that features such as low population density, high vaccination uptake, a low mean household size, and a low rate of negative LFD test reporting are protective. We note that for vaccination uptake, the protective value at zero is likely an artefact arising from cohorts with ages 0–9 that were not eligible.

The effects for many variables associated with social deprivation such as the ratio of working age people with no qualifications and the rate of income deprivation (see Section B.2 in S1 Text for full descriptions) are weaker. This is consistent with the small degree of deprivation-level variation seen in Fig 1.

The directionality of the ALEs remain broadly consistent across both waves. Some risk factors were more pronounced in the Delta model, including in mean hosehold size, population density and the proportion of individuals belonging to a black or minority ethnicity. Conversely, cohorts with very high student populations were associated more strongly with high case rates in the Omicron fit.

## 3 Discussion

Scotland's programme of free community testing was an invaluable tool for tracking the spread of COVID-19 infection up to early 2022. With the ending of detailed surveillance since, it is more difficult to monitor the precise patterns of infection amongst the population and how that will evolve over time, especially with respect to different variants.

The aim of this study was to compare the patterns of cases across two waves of COVID-19 in Scotland in 2021, during which non pharmaceutical interventions (NPIs) were being relaxed but testing remained mandatory and a mass vaccination rollout was in progress. We analysed the distribution of cases during the B.1.617.2 "Delta" wave from May 2021, and the B.1.1.529 "Omicron" wave from November 2021. We have shown that case heterogeneity was associated with broad factors such as age structure and residual immunity from earlier cases,

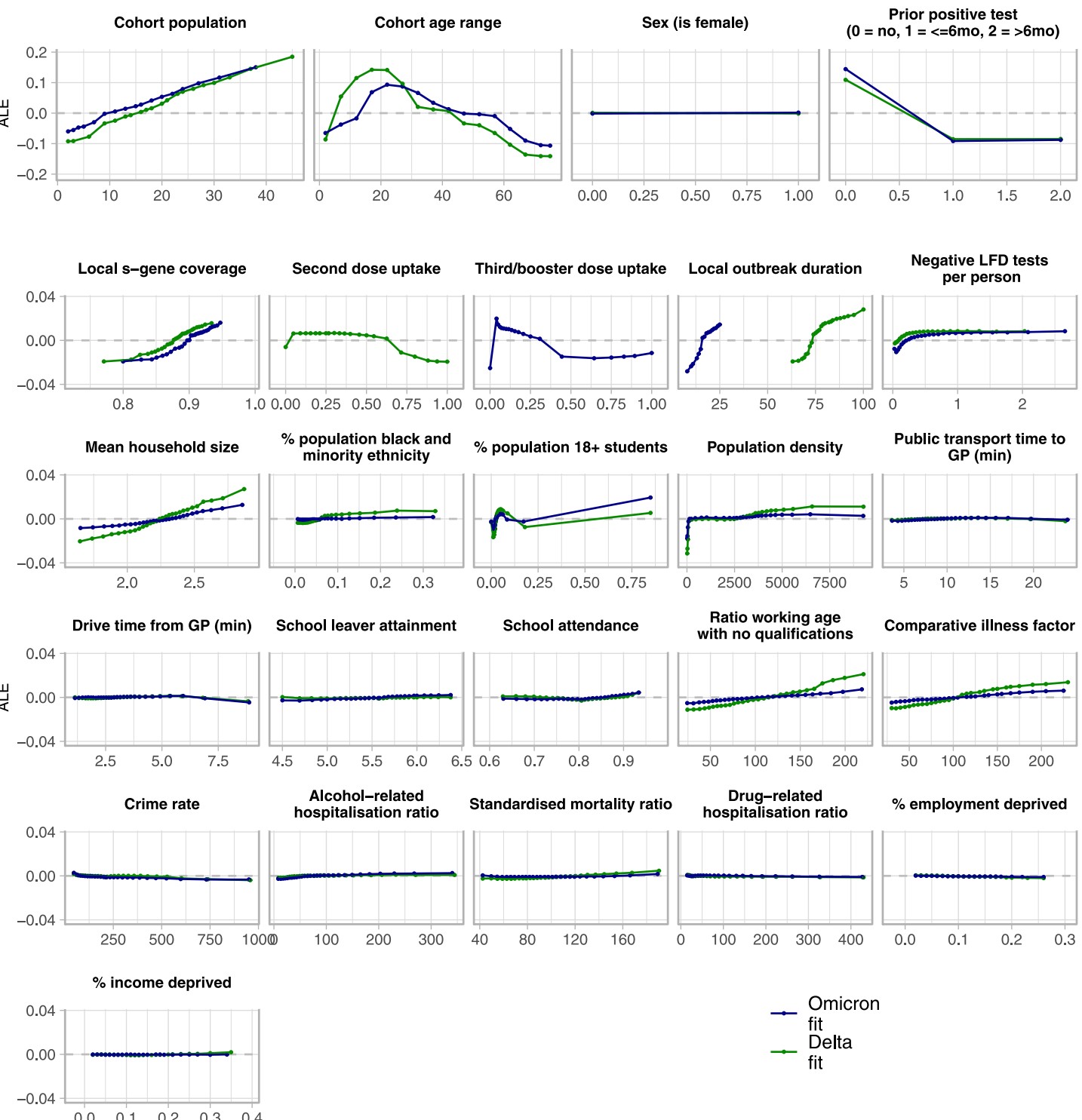

**Fig 4. Accumulated local effects across all explanatory variables.** For each variable, the *x*-axis represents the range of values of that variable in the data, and the *y*-axis (note scale differences for *population*, *age*, *sex* and *prior case status*) is the ALE for that variable value. The overall magnitude of the ALE represents the relative size of the effect.

but also with factors relating to testing, vaccination, geography and demographics. Despite differences in the severity of interventions in place, time of year, vaccination uptake and virus phenotype, these risk factors remain broadly consistent across both waves.

Our models accurately capture the case distributions (Fig 1). However, not all variation is explained, and residual autocorrelation persists at <5km scales (Fig 3). A reason for this may be that our model is not informed by mobility, thus explicit links between communities are not known to the model. We also do not include meteorological data (such as in e.g. [33]). This could have explained further variation as our waves occur in different seasons, where the characteristic routes of transmission may have differed. Last, the fit cases are time-aggregated, and therefore do not account for changes in risk factors *during* each wave.

The inclusion of the *local outbreak duration* for each DZ (the time the first case was detected in the DZs wider intermediate zone, typically containing 4–6 DZs) accounts in part for local interactions between neighbouring communities, in the absence of explicit mobility data. A weakness of this is that the local outbreak duration correlates with the total number of cases, given the relatively short periods studied. We suspect this is less influential in the Omicron model where geographical spread was more rapid. The regression models applied here may be better suited to scenarios where an infectious disease is already well established in the population. For future analyses on cases at the very beginning of an outbreak with fewer cases, this approach may be adapted to instead fit case rates per day, from when the first case was identified locally.

## 3.1 Risk factors

We presented the accumulated local effects (Fig 4), revealing broad indicators for higher or lower case rates, and how they changed between waves. It is difficult to fully disentangle whether a difference was caused by a change in control measures, or a change in virus strain. Nonetheless, our analyses provide some important insights.

To begin, high mean household size emerges as a risk factor, consistent with the high secondary attack rates for SARS-CoV-2 [40, 41], and increased risk of inter-household transmission relative to contacts outside of the home [42]. That this, and high population density are both stronger risk factors for Delta may reflect the stronger NPIs at this tme increasing the proportion of within-DZ or within-household transmissions.

High vaccine uptake (amongst those eligible) is also protective, more so with Delta, consistent with higher rates of immune breakthrough with the Omicron variant as compared to Delta [43–45]. We do not know the specific vaccination status of those in the test data, however, and linked data may show a stronger protective effect.

For Delta, a high proportion of individuals of black and minority ethnicity is a stronger risk factor. In the UK, this is also a risk factor for severe COVID-19 outcomes [46–48] but without detailed, linked data, it is difficult to firmly establish drivers for a *heightened* risk during the Delta wave. Differences may emerge from known variations in vaccination uptake [49] and occupation [50] (thus ability to work from home or effectively physical distance), and the relative impacts of those factors changing across the two waves.

Finally, living in a deprived community was suggested from early on [51] and has since also emerged as a risk factor for *severe* COVID-19 disease [52–57]. However, the corresponding ALEs for the variables associated with deprivation are small. Deprivation effects may be captured by proxy with other variables that correlate with deprivation such as age [58] and vaccine uptake [59, 60].

### 3.2 Testing frequency

The low case rate variation with deprivation (Fig 1) contrasts with observed inequalities over severe outcomes [22–25, 61], suggesting that those living in more deprived communities experience a higher inherent case-hospitalisation rate. We suspect that a lower proportion of case *ascertainment*, however, may also be a factor.

An important and unique variable in our model is the rate at which *negative* LFD tests were reported throughout the period. We found high rates of *negative* test reporting to be a *risk* factor. This suggests a variation in case ascertainment across different demographics, which may in turn lead to skews in the observed case distribution [22, 62, 63].

Further work (Fig H and Table A in S1 Text) shows that up to February 2023, the rate of LFD testing and positivity varied substantially across deprivation (quintile 1: 3.6 tests/person, 4.61% positive; quintile 5: 6.7 tests/person, 3.57% positive) as well as sex (M: 3.7 tests/person, 4.82% positive; F: 7.0 tests/person 3.30% positive). If demographic differences in testing behaviour correspond to differences in case ascertainment, the profile of all infections may then be biased from reported cases, and testing rates may be obscuring the true patterns of infection over sex and deprivation.

In addition, the magnitude of the risk factor (as seen in the ALE, Fig 4) plateaus beyond a certain rate ($> \sim 1$ test/person in each period). This hints at a deeper relationship between true incidence, the frequency of testing (and whom amongst the population is taking those tests), and the proportion of infections that are ascertained.

Our model is unique in including negative test reporting, and has revealed strong differences between different demographics that may bias the profile of cases. Beyond the work presented here, further analysess of reported cases need to be considered with these strong skews in testing behaviour in mind.

### 3.3 Conclusion

The COVID-19 data studied here are remarkable in terms of volume and resolution, and has allowed us to assess a national-level epidemic at extremely fine scale. However, regardless of resolution, cases only partially represent the full underlying pattern of infection. Variations in testing frequency and known trends in severe outcomes suggest that the distribution of infections may have been very different to that of reported cases. By incorporating trends on cases, testing behaviour, and severe outcomes more closely linked to infection (hospitalisation, ICU admission and mortality), it may be possible to build a much more comprehensive retrospective picture of how infections were distributed amongst the population.

Importantly, while our access to such finely-grained data was exceptional, it can be expected that such data are likely to become more common in the future, and may become available in real time. As such, our demonstration of the utility of such data points the way to an important approach to improving data analysis supporting control policy response to infectious disease emergencies in the future.

## 4 Data and methods

### 4.1 Preparation of case data

We use COVID-19 testing data from Public Health Scotland's *electronic Data Research and Innovation Service* (eDRIS) system, dated from July 14th 2022. The data include individual tests by type (polymerase chain reaction (PCR) or rapid lateral flow device (LFD)), test result (positive, negative, void, inconclusive), test date, S-gene test result if known (positive, dropout, inconclusive), age, sex, and residing data zone (*DZ*, a census area typically comprising 500–

1,000 individuals). De-identified IDs link repeat tests by the same individual. We reduce the raw test data to cases by removing duplicate tests by the same individual within 60 days (taking the date of the first PCR positive as the case date, or the first LFD in the absence of any PCR). These metadata—in particular the DZ, specifying location to within an area as small as 0.1km$^2$ in densely populated areas—therefore identify cases at a fine spatio-temporal scale. Data on vaccine administrations are also provided by eDRIS.

This analysis considers the BA.1 sub-variant of the Omicron lineage only. The sub-variant BA.2/B.1.1.529.2 later replaced BA.1, becoming dominant in Scotland from around 25[th] February 2022. This variant, like Delta, has an S-gene positive test signature. However by the end of the period studied the BA.2 variant was only being identified in fewer than 1% of fully sequenced cases in the UK [64], and here we assume all remaining S-gene positive cases to be Delta.

Prior to January 6[th] 2022 in Scotland, positive LFD tests (typically taken at home) required PCR confirmation. Approximately 90% of cases in this period have a definitive S-gene result. A policy change then dropped this PCR requirement [65], after which cases with S-gene results fell to about 50% by February 2022 (per eDRIS data).

For Omicron cases, we gather from the data S-gene dropout cases between 15[th] November 2021 and 6[th] January 2022, and for the Delta outbreak, S-gene positive cases between 1[st] May and 7[th] September 2021 (choosing this end date to have a similar number of cases in each set). We exclude cases that have a different, or no S-gene result.

Using the linked historical tests, we label cases based on whether the individual had either: never tested positive before; had tested positive in the last six months prior to the start of that wave, or; last tested positive over six months prior to the start of that wave. We denote this the *prior case status*, as a proxy for infection-based immunity.

Finally to prepare the cases data to be fit, we group individuals that have the same age range, sex, residing datazone, and *prior case status*, terming these subsets of individuals *cohorts*. As an illustrative example, a cohort may be a population of 38 males aged between 50–54 residing in a given datazone "X", that have never tested positive for COVID-19 before, among whom 9 Omicron COVID-19 cases were identified. This is the highest practical resolution we can acheive using the eDRIS case data, and our model (Section 4.3) fits case counts at this resolution.

## 4.2 Time series analysis

### 4.2.1 Time-dependent reproduction number.
The time-dependent reproduction number $R_i$ is the average number of forward infections caused by a person infected on day $t_i$. Define $n_j$ as the number of new infections on day $t_j$. These new infections came from individuals infected on days on, or prior to $t_j$. Define $A_{ij}$ as the number of new infections on day $t_j$ *specifically* from those infected on day $t_i \leq t_j$:

$$A_{ij} = \frac{(n_i - \delta_{ij})P(t_j - t_i)}{\sum_{i' \leq j}(n_{i'} - \delta_{i'j})P(t_j - t_{i'})}\, n_j\,.$$

$P(\Delta t)$ is the probability of an individual passing on the infection, $\Delta t$ days after being infected. The presence of the Kronecker delta $\delta_{ij}$ excludes the possibility of infected individuals infecting themselves. The reproduction number $R_i$ is then the average total of infections generated over

all subsequent days [66]:

$$R_i \;\; = \frac{1}{n_i}\sum_{j\geq i} A_{ij} = \frac{1}{n_i}\sum_{j\geq i}\frac{n_j(n_i-\delta_{ij})P(t_j-t_i)}{\sum_{i'\leq j}(n_{i'}-\delta_{i'j})P(t_j-t_{i'})}\;\; .$$

We take $P(\Delta t)$ to be

$$P(\Delta t)\sim e^{-\lambda\Delta t}$$

with $\lambda^{-1}$ the mean infectious period. Individuals are equally infectious throughout the entire infection. In our calculations we estimate $1/\lambda = 6.26$ days, using the posterior mean duration of infectiousness obtained from the *SCoVMod* compartmental model (for more detail see Reference [57]).

As we estimate the infection reproduction number using the cases data, we implicitly assume that case ascertainment does not change over time, and does not account for the delay between infection, and registering a case.

In this work the reproductive number is measured at local authority level, the level at which the Scottish Government monitored and adjusted NPIs.

**4.2.2 Case doubling time.**   At the start of each wave we assume exponential growth of cases:

$$\text{new cases} \propto e^{rt}$$

where the gradient of a linear regression on log (new cases) against $t$ returns the growth rate $r$. The evolution of new cases an also be rewritten in terms of of a doubling time $t_D$:

$$\text{new cases} \propto 2^{t/t_D}$$

where $t_D = \frac{\log 2}{r}$.

## 4.3 Model

Our statistical model is designed to explain variation in COVID-19 case numbers as prepared in Section 4.1, and identify risk factors amongst a broad range of variables, using random forest regression. We fit models to the distribution of Delta and Omicron cases respectively, allowing for comparison of risk factors across the two waves.

**4.3.1 Explanatory variables.**   We include demographic factors (population, age, sex, ethnicity, student population), COVID-19 related factors (testing volume, prior case status, vaccination uptake), geography (local population density and transport time to public services to serve as proxies for connectivity and geographic remoteness), as well as deprivation. Data on deprivation are taken from the *Scottish Indices of Multiple Deprivation* (SIMD) [67]. The SIMD ranks DZs in Scotland by "multiple" deprivation, incorporating measures relating to local health, housing, geographic access, employment, income, crime, and education. In our model we use the raw measures of deprivation as explanatory variables. To account for local spread of infection between neighbourhoods that are geographically close to one another, we include an *local outbreak duration* parameter, which specifies the date at which the *first* case of the variant was identified at the *intermediate zone* (IZ, an administrative area containing of order 4–6 DZs).

A comprehensive description of all individual variables used is given in Section B.2 in S1 Text.

**4.3.2 Random forest model.**   We use random forest regression [68] on the distribution of COVID-19 cases, as it allows us to fit the distribution without specifying any prior analytical

relation between the outcome variable (cases) and any of the explanatory variables, which may themselves be correlated. We fit the time-aggregated case distribution in *R* (version 4.1.0) [69], using the *randomForest* package [70] (version 4.6–14).

We fit the outcome variable $\sqrt{cases + 1}$ at cohort level (with a *cohort* defined in Section 4.1). The fit number of cases at other scales (such as DZ level) is then an aggregation of cases from their constituent cohorts.

We extract two metrics for variable importance from the *randomForest* function output: the node purity (a measure of how effective variables are at partitioning cohorts with differing numbers of cases in the tree), and the loss of model accuracy on effective removal of that variable from the model.

Model hyperparameters were chosen manually so as to maximise the variance explained by a subset of the data not used to fit the model. Full hyperparameter specification is included in Section B.1 in S1 Text. The model specifications for fitting the Omicron and Delta waves are identical with one exception: for the Omicron model, third/booster dose uptake is used, whereas for Delta, second dose uptake is used (third/booster doses were only administered later; see Section B.3 in S1 Text for further details).

In addition to the full model, we fit for each of Omicron and Delta three "reduced" models, under equivalent hyperparameters to the full model and the same cohort structure, but informed only by population, and one of: age; the relative deprivation of the residing DZ, as defined by the overall SIMD deprivation *rank* [71], and; population density. These outputs illustrate how effective these variables are at alone at explaining case variation, relative to our full model.

## 4.4 Accumulated local effects

To identify risk factors amongst the explanatory variables used to inform the model, we calculate the *accumulated local effects* (ALEs) of each variable. The ALEs describe how the model fit value changes, in response to changing one variable value in isolation, averaged over many different entries in the data [72]. In this context, ALEs indicate whether a variable value is associated with fewer or more cases in general over the data. If the ALE is greater than zero, the fit cases generally increases given that variable value.

## 4.5 Moran's I autocorrelation statistic

To probe geographical variation in cases *not* explained by the model, we measure the Moran's I autocorrelation [73, 74] on the residuals (the difference between the data and fit value), relating to their physical location. We compare local DZ-aggregated residuals over physical distances (from 1–100km), as well as network distance (number of nearest neighbours apart). For a set of $N$ residuals $y_i$, the Moran's I is a measure of autocorrelation:

$$I = \frac{N}{\sum_{i=1}^{N}\sum_{j=1}^{N} w_{ij}} \frac{\sum_{i=1}^{N}\sum_{j=1}^{N} w_{i,j}(y_i - \bar{y})(y_j - \bar{y})}{\sum_{i=1}^{N} (y_i - \bar{y})^2}$$

with $\bar{y}$ the mean of all residuals, and $w_{i,j}$ is an associated *weight* of the pair of observations $(i, j)$, with $w_{i,i} = 0$. To measure the autocorrelation between residuals within a separation $d$ (either a physical or network-based distance) of one another, we set $w_{i,j} = 1$ if $\text{dist}(i, j) \leq d$, and 0 otherwise. Fully correlated residuals would have $I = 1$, whereas $I = 0$ would indicate no correlation.

This measure characterises how effective our models are at explaining geographical variation, and with different distances $d$ shows over what length scales residual autocorrelation persists.

## Supporting information

**S1 Text.** **A**. Supplementary plots for the time evolution of cases across the Delta and Omicron waves. **B**. Additional methodology details; hyperparameter selection, detailed description of all explanatory variables. **C**. Map view of population distribution of Scotland, and model residuals for Omicron model. **D**. Plots for explanatory variable Importance; node purity, accuracy loss on variable permutation. **E**. Additional details on lateral flow testing frequency, broken down by sex and deprivation quintile.
(PDF)

## Acknowledgments

We thank Public Health Scotland's *electronic Data Research and Innovation Service* (eDRIS) for the provision of COVID-19 testing, vaccination and severe outcomes data. We also thank the reviewers for their feedback and suggestions, which has led to improvement of the article.

## Author Contributions

**Conceptualization:** Rowland R. Kao.

**Formal analysis:** Anthony J. Wood, Aeron R. Sanchez, Paul R. Bessell, Rebecca Wightman.

**Funding acquisition:** Rowland R. Kao.

**Investigation:** Anthony J. Wood.

**Methodology:** Anthony J. Wood, Rowland R. Kao.

**Project administration:** Rowland R. Kao.

**Supervision:** Rowland R. Kao.

**Visualization:** Anthony J. Wood.

**Writing – original draft:** Anthony J. Wood, Rowland R. Kao.

**Writing – review & editing:** Anthony J. Wood, Rowland R. Kao.

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
