## [Decision Letter · Decision Letter 0]

12 Jul 2023

Dear Professor Kao,

Thank you very much for submitting your manuscript "Assessing the importance of demographic risk factors across two waves of SARS-CoV-2 using fine-scale case data" for consideration at PLOS Computational Biology.

As with all papers reviewed by the journal, your manuscript was reviewed by members of the editorial board and by several independent reviewers. In light of the reviews (below this email), we would like to invite the resubmission of a significantly-revised version that takes into account the reviewers' comments.

We cannot make any decision about publication until we have seen the revised manuscript and your response to the reviewers' comments. Your revised manuscript is also likely to be sent to reviewers for further evaluation.

Sincerely,

Claudio José Struchiner, M.D., Sc.D.

Academic Editor

PLOS Computational Biology

Virginia Pitzer

Section Editor

PLOS Computational Biology

Reviewer's Responses to Questions

**Comments to the Authors:**

Reviewer #1: This manuscript describes an analysis of COVID-19 testing data aggregated across local geographical units in Scotland, between two time periods with different dominant variants. Overall I feel that the manuscript lacks focus - it is not clear what the aim of the work is and how this fits into the existing literature. The methods section does not give sufficient information to understand what analyses were conducted, how they were conducted, and to address what questions. Some conclusions are drawn which are not supported by what is presented in the paper, or by other cited sources. The conclusions which are related to these specific results don’t appear to add to what is already known on the subject.

Introduction

Little mention of related literature

The objective is not clear. Background paragraphs are broad and non-specific to this analysis. Is the interest in identifying risk factors for transmission/infection/mortality?

Mentions heterogeneity and “finely-grained data” and I assume the authors mean with respect to space, but this is not explicit.

The authors should better describe the existing literature that has explored these kind of data/patterns, and highlight how what you’re doing is different/complementary.

Could give more context about the pandemic in Scotland specifically.

The introduction mostly describes the analysis approach, rather than providing relevant context and justifying the question to be addressed/the approach taken to address it.

Data and Methods

Unclear why the data were split into “cohorts”.

What is the time scale for last testing positive? By month, week?

If observations are already split into cohorts based on age/sex, is it appropriate to then use age/sex again as explanatory variables? Why necessary to do both?

There should be a clear model specification included here, explicitly defining the outcome(s) and predictors.

The analysis approach should be described fully, referring to all variations of the model that were fit (results mentions a “full” model as well as several “univariate” but these are not defined).

What was different between the “case distribution” and “spatial” analysis?

Descriptive analyses should also be explained here (the results section talks about “regression on log_10(cases)”, doubling times, R_t, a hypothesis test of age between variants).

Reading this section it isn’t apparent what question is being addressed. Is the aim to compare between periods or variants? Or to compare explanatory power of different variables?

Why the need to have similar number of cases in each time period?

Are you excluding cases caused by other variants in the same periods?

Results

Meaning of S-gene dropout/positive and a timeline of the different variants should be briefly explained.

If spatial variation is a focus then maps of the area/outcomes should be presented.

Both data zones and local authorities are discussed – why move between the two?

Figure 1 shows cases and hospitalisations – is the analysis repeated for both? I don’t think this has been mentioned.

Reference to calculating Moran’s I which is not described in the methods.

The results section generally includes a lot of description of methods which should be in the methods section.

Discussion

The authors reference on multiple occasions covid “circulating in high volume” – I’m not sure I know what you mean by this.

There is a lot of literature on risk factors for infection/mortality throughout the pandemic which has not been referenced. I don’t agree that there is “increasing uncertainty” in who is at highest risk – as information has accumulated over time our understanding has improved.

“Any differences as were observed are as likely to be explained by the differences in imposed interventions as they are due to differences in the virus strains.” – this conclusion is not supposed by anything presented here.

Reference to “individual level” drivers but as far as I can tell the model was fit to aggregated observations (“cohorts”)

“local outbreak duration parameter” – this hasn’t been mentioned anywhere prior as far as I can tell

Reference to “sampling” of cohorts but methods just says that cohorts were defined by grouping observations, not sampling from some wider pool.

Reviewer #2: The manuscript "Assessing the importance of demographic risk factors across two waves of SARS-CoV-2 using fine-scale case data" by Wood et al. provides a comprehensive analysis of regional variations in SARS-CoV-2 spread using fine-scaled data on cases combined with various epidemiological and socio-economic factors. While none of the individual results presented in the paper are new by themselves, the study gives a quite comprehensive picture that allows one to better understand the importance of some of the factors in relation to other risk factors. The authors achieve a remarkable fit between their model and data, suggesting that the risk factors covered in these factors indeed capture most of the variation in regional spread. I just have a few minor comments aimed at improving the clarity and interpretability of the results.

-) Overall, figures are sometimes hard to read due to the extremely small font size.

-) Regarding Fig. 2 and its caption, I'm not sure that the meaning of the results conveyed here are easily accessible to non-technical experts. If I understand it correctly, the message is that close-by regions (in two different senses) tend to show similar deviations with respect to the model, meaning that the unexplained variance is (at least in parts) due to regional influences? Maybe something along these lines can be stated more clearly. I was also wondering whether such an effect can also be seen by coloring the points in a scatter plot of data vs full model in a way that encodes administrative divisions/regions, or showing centroids of points belonging to the same region and some measure of their spread?

-) The differences in scales in Fig. 4 are a bit unfortunate but I understand their necessity. To make this clearer, one could maybe put population, age, sex, prior test on the same scale within one row and a bit detached form that the other variables, also all on the same scale. Furthermore, some of the factors with small ALEs that are also not really discussed at any point in the manuscript could be moved to the SI entirely.

-) Speaking of which, what was the rationale behind the inclusion of some of the risk factors that show no ALEs and are not really discussed? What does this add to the study? Also negative results should be discussed.

-) Some discussion on what are possible risk factors that account for the unexplained variance might be informative. Potential differences in non-pharmaceutical interventions, mobility and meteorological factors come to my mind based on previous literature:

https://doi.org/10.1016/S1473-3099(20)30553-3

https://doi.org/10.1371/journal.pcbi.1009973

https://doi.org/10.1073/pnas.2019284118

Reviewer #3: The Authors used a random forest model to assess the impact of various factors such as age, sex, prior infections and deprivation status on the risk of COVID-19 infection (variants Delta and Omicron). A definite strong point of the work is the unique dataset, comprising detailed, high-resolution spatial and demographic information about cases and hospitalizations in Scotland. The study is well-constructed, the hyperparameters are set to avoid overfitting, and the models are validated on an independent set. The results show that the risk factors were fairly consistent between the variants, which is an interesting observation.

Although the majority of features used in the model can be considered independent variables, my one concern is that the local outbreak duration was chosen as a proxy for spatial relationships between data zones. Particularly in densely populated areas, where data zones are likely more arbitrary and highly similar, the intermediate zone can be indicative of the duration in the considered zone itself. Since the outbreak duration depends on the number of cases, it is a derivative of the response variable, rather than a predictor. Indeed, both the ALE and model accuracy loss analyses showed high importance of this variable. Furthermore, since this information can only be obtained after the outbreak, inclusion of this variable limits the applicability of the proposed model for potential future COVID-19 waves.

While I found the article a compelling read, I would also like to point out several technical issues:

- In line 69, please specify the year for both dates, since it is not the same.

- In Fig. 1, please extend the figure description for a clearer perception - please specify what the numbers are relative to, perhaps include the N for the whole population. Moreover, the layout for the prior cases panel is different from the rest. Were no previously infected individuals hospitalized?

- In Fig. 2 and Supplementary Fig A.3 (B), the x axis tick labels overlap slightly.

- In line 100, please replace " 'univariate' " with a more precise term - perhaps "reduced".

- Particularly in the Results section, the descriptions tend to be slightly chaotic - for example, lines 169-174 would be more fitting for the "Detailed case distribution analysis" subsection.

- From the description, it seems that the model hyperparameters were set arbitrarily. Please avoid using the word "optimised", "maximises" etc. unless an optimisation procedure was performed. If different hyperparameter values were indeed tested, the results would be a valuable addition.

- For non-standard abbreviations (eDRIS, DZ) please include the explanation not only in the Materials and Methods section, but also at the first use - it would improve readability.

- Please ensure that all of the references adhere to the required format and are consistent - for example, in reference 28, the Office for National Statistics is mentioned after the title, and in reference 31 before, as "for National Statistics, O". For online resources, please include the access date, or, in case of R packages, the version number.

Reviewer #4: The authors of the article, among others, built a model using random forests to predict the distribution of cases in Scotland. They applied modeling to two successive waves (Delta, Omicron) of the Covid-19 pandemic. They concluded that the distribution of cases could, to some extent, be explained by independent variables such as age, deprivation and population density. In addition, they indicated that the detected relationships remain largely unchanged regardless of which wave of the Covid-19 pandemic was analyzed.

The presented results look very interesting. Unfortunately, the authors did not take care to explain in detail how the predictive model was built and what data it was supplied with.

Major comments.

The authors write that they consider the so-called geographical data zones (DZs) covering about 1000 inhabitants each. As a reviewer, I understand this to mean that from a machine learning (or statistical) point of view, one data zone constitutes one pattern (observation). This is suggested by Figure 2, in the description of which it is indicated that one point corresponds to one DZ. The problem is that the authors have not clearly described what the explained and explanatory variables are. I do not understand what is on the axes of the graphs shown in Figure 1. The axes are marked with the word "Cases" - what does it mean? Number of Covid-19 cases in one data zone? The range on these axes is 1-100. Is it a percentage number of cases in one DZ?

A more serious problem is the explanatory variables. While deprivation or population density, for example, are obvious, it is not entirely clear what the "Age" variable means for a given data zone. If there are about 1000 inhabitants in one DZ, does "Age" mean average age or median age? If so, it seems that the variable "Age" defined in this way should be more or less constant for different areas of Scotland. Given this, it is strange that a relatively good match was achieved for "Age", comparable to "Deprivation" or "Density". It is possible that this has something to do with the "Accumulated local effects" described in section 2.2.1. If so, it should be clearly stated.

Minor comments.

Does Figure 2 show the results for the training set or the test set?

The abbreviation "DZ" was not explained anywhere and as a reviewer I had to guess it.

The authors wrote in the abstract that One DZ covered about 1000 inhabitants. And elsewhere, in the caption of Figure A4, it says 500-1000. What is the truth?

It would be interesting to build a model for the first Delta wave of the Covid-19 pandemic and use it to predict the second wave of Omicron.

**Have the authors made all data and (if applicable) computational code underlying the findings in their manuscript fully available?**

Reviewer #1: **No: **Data are said to be not publicly available, however justification as to why they cannot be made available isn't given. No suggestion is given as to how the data may be acquired.

Reviewer #2: **No: **Data is not publicly available as described in the data availability statement but a procedure to obtain access the the data is described.

Reviewer #3: **No: **As indicated in the manuscript, a part of the data used for the analysis was only provided for the Authors privately under a data sharing agreement by a national government agency, and as such is not available for the general public.

Reviewer #4: Yes

PLOS authors have the option to publish the peer review history of their article (what does this mean?). If published, this will include your full peer review and any attached files.

Reviewer #1: No

Reviewer #2: No

Reviewer #3: No

Reviewer #4: No
---

## [Decision Letter · Decision Letter 1]

17 Oct 2023

Dear Professor Kao,

We are pleased to inform you that your manuscript 'Assessing the importance of demographic risk factors across two waves of SARS-CoV-2 using fine-scale case data' has been provisionally accepted for publication in PLOS Computational Biology.

Best regards,

Claudio José Struchiner, M.D., Sc.D.

Academic Editor

PLOS Computational Biology

Virginia Pitzer

Section Editor

PLOS Computational Biology

Reviewer's Responses to Questions

**Comments to the Authors:**

Reviewer #1: I am pleased see that revised submission is significantly improved and I'd like to thank the authors for their thorough and thoughtful responses to all reviewers' comments. My queries have been addressed and I'd now be happy to recommend the article for publication.

Reviewer #2: The authors made a great effort to address the comments and have made extensive changes to further improve the mansucript. I recommend accepting the manuscript.

Reviewer #3: The structure of the manuscript was substantially revised and some information was moved from the supplementary file to the main text, which was beneficial for readability and clarity of the main objective. The figures are much clearer and the editing was improved.

I also appreciate the explanation regarding hyperparameter optimization, suggesting that the presented results are for the selected model in a validation procedure.

Reviewer #4: As a reviewer, I am satisfied with the current form of the manuscript. The authors responded to all my comments. In my opinion, the article in its current form is suitable for publication.

**Have the authors made all data and (if applicable) computational code underlying the findings in their manuscript fully available?**

Reviewer #1: **No: **The reason for this is explained and a contact for enquiries about the data is given.

Reviewer #2: **No: **Population health data has been used which is only accessible under specific conditions. Code is available.

Reviewer #3: **No: **As explained by the Authors, parts of the data were provided to them under a data sharing agreement by a government institution and are not available.

Reviewer #4: Yes

PLOS authors have the option to publish the peer review history of their article (what does this mean?). If published, this will include your full peer review and any attached files.

Reviewer #1: No

Reviewer #2: No

Reviewer #3: No

Reviewer #4: No

---

## [Editor Report · Acceptance letter]

16 Nov 2023

PCOMPBIOL-D-23-00521R1 

Assessing the importance of demographic risk factors across two waves of SARS-CoV-2 using fine-scale case data

Dear Dr Kao,

I am pleased to inform you that your manuscript has been formally accepted for publication in PLOS Computational Biology. Your manuscript is now with our production department and you will be notified of the publication date in due course.

With kind regards,

Bernadett Koltai
